# Alpha-synuclein and tau are abundantly expressed in the ENS of the human appendix and monkey cecum

Alexandra D. Zinnen[1], Jonathan Vichich[1], Jeanette M. Metzger[1], Julia C. Gambardella[1,2], Viktoriya Bondarenko[1], Heather A. Simmons[1], Marina E. Emborg[1,2,3]*

1 Wisconsin National Primate Research Center, University of Wisconsin-Madison, Madison, Wisconsin, United States of America, 2 Cellular and Molecular Pathology Graduate Program, University of Wisconsin-Madison, Madison, Wisconsin, United States of America, 3 Department of Medical Physics, University of Wisconsin-Madison, Madison, Wisconsin, United States of America

* emborg@primate.wisc.edu

**Data Availability Statement:** All relevant data are within the paper and its Supporting Information file (S1 Table).

## Abstract

α-Synuclein (α-syn) proteinopathy in the neurons of the Enteric Nervous System (ENS) is proposed to have a critical role in Parkinson's disease (PD) onset and progression. Interestingly, the ENS of the human appendix harbors abundant α-syn and appendectomy has been linked to a decreased risk and delayed onset of PD, suggesting that the appendix may influence PD pathology. Common marmosets and rhesus macaques lack a distinct appendix (a narrow closed-end appendage with a distinct change in diameter at the junction with the cecum), yet the cecal microanatomy of these monkeys is similar to the human appendix. Sections of human appendix (n = 3) and ceca from common marmosets (n = 4) and rhesus macaques (n = 3) were evaluated to shed light on the microanatomy and the expression of PD-related proteins. Analysis confirmed that the human appendix and marmoset and rhesus ceca present thick walls comprised of serosa, muscularis externa, submucosa, and mucosa plus abundant lymphoid tissue. Across all three species, the myenteric plexus of the ENS was located within the muscularis externa with nerve fibers innervating all layers of the appendix/ceca. Expression of α-syn and tau in the appendix/cecum was present within myenteric ganglia and along nerve fibers of the muscularis externa and mucosa in all species. In the myenteric ganglia α-syn, p-α-syn, tau and p-tau immunoreactivities (ir) were not significantly different across species. The percent area above threshold of α-syn-ir and tau-ir in the nerve fibers of the muscularis externa and mucosa were greater in the human appendix than in the NHP ceca (α-syn-ir $p<0.05$; tau-ir $p<0.05$). Overall, this study provides critical translational evidence that the common marmoset and rhesus macaque ceca are remarkably similar to the human appendix and, thus, that these NHP species are suitable for studying the development of PD linked to α-syn and tau pathological changes in the ENS.

**Funding:** This research was funded by grants from the National Institute of Health [NIH P51OD011106; https://orip.nih.gov/] (ME), the Parkinson's Foundation [PF-APDA-SFW-1922; https://www.parkinson.org/] (AZ) and Trewartha Senior Honors Thesis Research Award from the University of Wisconsin–Madison [https://wisc.academicworks.com/opportunities/48209] (AZ). The funders had no role in study design, data collection and analysis, decision to publish, or preparation of the manuscript.

**Competing interests:** The authors have declared that no competing interests exist.

## Introduction

The presence of a vermiform appendix is considered a defining characteristic of the superfamily Hominoidea, which is composed by humans and apes [1]. The vermiform appendix is a narrow diverticulum at the end of the cecum, with thick walls comprised of serosa, muscularis externa, submucosa and mucosa plus abundant lymphoid tissue organized in follicles, termed Peyer's patches [1]. Regarded for many years as a vestigial organ [1], the appendix is now known to have a role in the immune system by stimulating development of gut associated lymphoid tissue (GALT) in the gastrointestinal (GI) tract [2].

An unexpected connection between the appendix and Parkinson's disease (PD) has recently emerged. Based on data from the Swedish National Patient Registry, two separate studies found that appendectomy was associated with lower risk of PD in all populations [3] and in residents of rural areas [4]. The Parkinson's Progression Markers Initiative database related appendectomy to a delayed onset of PD [4]. These findings suggest that the appendix influences events leading to clinical onset of PD, which could help understand why GI symptoms, such as constipation, are common in PD and frequently predate the onset of typical PD motor symptoms [5].

Proteinopathy is proposed to be the link between the appendix and PD. α-Synuclein (α-syn) is a presynaptic protein associated with neurotransmitter vesicle release and is ubiquitously expressed in neurons of the central and peripheral nervous system [6]. In non-pathological conditions, α-syn is found in a soluble form [7]. However, in patients with PD, α-syn phosphorylates, aggregates and becomes the main component of Lewy bodies (LBs), a pathological hallmark of PD [7]. Tau is a member of the microtubule-associated protein family [8, 9] and, in normal conditions, plays a role in stabilizing neuronal microtubules [10]. In PD, hyperphosphorylated tau is a component of LBs and it also forms intraneuronal neurofibrillary tangles [11, 12]. The presence of both α-syn and tau in LBs suggests an interaction between the two proteins leading to PD pathology [8, 9].

Braak's hypothesis proposes that sporadic PD starts in the GI tract and the enteric nervous system (ENS) after an unknown environmental trigger is ingested, inducing LB formation and spreading of proteinopathy from the ENS to the CNS [13]. Phosphorylated α-syn and tau have been identified in the ENS of the colon, as well as in the submandibular gland and skin of PD patients [14–17]. Aggregated and phosphorylated α-syn were found in vermiform appendix tissue samples from PD patients and, surprisingly, also from healthy subjects [4, 18, 19]. Further, α-syn-positive nerve fibers were more abundant in the vermiform appendix than in the stomach, ileum, or colon, suggesting that the vermiform appendix is an attractive candidate organ for the initiation of enteric α-syn aggregation [18].

Common marmosets (*Callithrix jacchus*) and rhesus macaques (*Macaca mulatta*) are nonhuman primates (NHPs) from the New and Old World, respectively. They are widely used in biomedical research, including for modeling PD [20]. The cecal anatomy of these NHPs closely resembles the human appendix [1], but little is known regarding its α-syn and tau expression. If indeed the appendix has a role in PD proteinopathy and disease onset, differences between human and NHP species may preclude their use for disease modeling and/or shed light on disease onset linked to α-syn and tau pathological changes. Here, we report our characterization and comparative analysis of α-syn and tau expression in the cecum of common marmoset and rhesus monkeys and the appendix of humans. We aimed to assess the validity of these NHP species for modeling PD pathological processes associated to GI proteinopathy.

## Material and methods

### Ethics statement

The present study was performed in strict accordance with the recommendations in the National Research Council Guide for the Care and Use of Laboratory Animals (8[th] edition, 2011) in an AAALAC accredited facility, the Wisconsin National Primate Research Center (WNPRC) at the University of Wisconsin-Madison. Experimental procedures were approved by the Institutional Animal Care and Use Committee (IACUC) at the University of Wisconsin-Madison (original experimental protocol wprc00). All efforts were made to minimize the number of animals used and to ameliorate any distress.

Fully de-identified, anonymized human tissue samples were obtained from the Translational Research Initiative (TRIP) Lab in the department of Pathology and Laboratory Medicine at the University of Wisconsin-Madison (appendix), the Translational Science Biocore Biobank (TSB Biobank) at the University of Wisconsin Carbone Cancer Center (appendix), and the Wisconsin Alzheimer's Disease Research Center Brain Bank at the University of Wisconsin-Madison (brain), following documented review and approval from the Institutional Review Board.

### Tissues

Human appendix tissue samples (n = 3) and cecum tissue samples from adult common marmosets (n = 4) and rhesus macaques (n = 3) were used in this study (see Table 1 for demographics). Criteria for the eligibility of the appendix and cecum tissue sample were: 1) diagnosis of healthy morphology by a board certified veterinary anatomic pathologist (H.A.S.) and 2) integrity of the anatomic layers of the cecum, including the mucosa, submucosa, muscularis mucosa and muscularis externa.

All ceca were collected and processed following previously published methods [21, 22]. Appendices and ceca were post-fixated in 4% paraformaldehyde, further preserved with 70% ethanol, 4mm tissue slabs were cut in the transverse plane, processed and embedded in paraffin. The human and NHP paraffin embedded tissues were cut on a standard rotary microtome in 5μm section thickness and mounted on positively charged slides.

### Immunohistochemistry

Immunohistochemistry against α-syn, phosphorylated α-syn (p-α-syn), tau and phosphorylated tau (p-tau) was performed on sections of appendix and cecum as previously described [21, 22]. Sections were deparaffinized and treated for heat antigen retrieval in a microwave for

**Table 1. Demographics of NHPs used in this study.**

| Subject ID | Age (yr) | Sex | Weight (kg) |
|---|---|---|---|
| **Common Marmoset** | | | |
| cj1 | 7.24 | Female | 0.31 |
| cj2 | 11.5 | Male | 0.41 |
| cj3 | 4.54 | Male | 0.42 |
| cj4 | 7.8 | Male | 0.38 |
| **Rhesus Macaque** | | | |
| rh1 | 13.6 | Male | 15.25 |
| rh2 | 9.6 | Male | 14.49 |
| rh3 | 10.9 | female | 9.7 |

6 minutes at 100% power followed by 3 minutes at 60% power and left to cool for 1 hour at room temperature. The sections were then washed and endogenous peroxidase activity blocked by incubation in a mix of 30% $H_2O_2$ and 60% methanol. Nonspecific binding sites were blocked with Super Block (ScyTek, Logan, UT) for 60 minutes at room temperature and then incubated overnight with primary antibody (Table 2). The primary antibody was diluted in blocking buffer plus 0.1% Triton-X. The sections were then incubated in an appropriate biotinylated secondary antibody (1:200), followed by avidin-biotin-complex peroxidase (VECTASTAIN Elite ABC HRP Kit, Vector Laboratories, Burlingame, CA) and visualized with a commercial 3,3'-diaminobenzidine (DAB) kit (Vector Laboratories, Burlingame, CA). Appendix and cecum sections were counterstained with Hematoxylin, dehydrated and coverslipped (Cytoseal mounting medium, Thermo Scientific, Waltham, MA, USA).

## Immunofluorescence

Triple-label immunofluorescence was performed to assess distribution of the proteins α-syn, p-α-syn, tau and p-tau in the ENS ganglia and nerve fibers identified by the panneuronal marker PGP9.5. Slides were deparaffinized and treated for heat antigen retrieval in a microwave for 6 mins at 100% power followed by 3 mins at 60% power and left to cool for 30 mins at room temperature. Tissue was incubated with 5% donkey serum and 2% BSA solution, followed by primary antibodies (Table 2) overnight at 4˚C. The sections were then incubated with Alexa Fluor-conjugated secondary antibody (1:1000) against the appropriate species and coverslipped with mounting medium with 4′,6-diamidino-2-phenylindole (DAPI) (Vector Laboratories, Burlingame, CA). Negative controls were performed in parallel by omitting the primary antibodies. Confocal images were obtained using a Nikon A1 confocal microscope (Tokyo, Japan).

## Anatomical measures of appendix and cecum

Dimensions of the overall anatomy of the appendix and cecum were obtained for comparison across species using NIH ImageJ software version 2.1.0. The diameter of appendix and cecum were calculated using the length function on images of whole tissue sections captured using an Epson 1640XL-GA high resolution digital scanner. All other measures were obtained from 1.25x magnification images acquired with a Zeiss Axioimager M2 microscope. The perimeter was calculated by drawing a region of interest (ROI) around each tissue section and applying the perimeter function. The area of tissue on the slide that contained all GI wall layers was

**Table 2. Antibody information used in general immunohistochemistry and immunofluorescence staining.**

| Antigen | Marker | Company | Species | Catalog no. | Lot no. | Dilution |
|---|---|---|---|---|---|---|
| General Immunohistochemistry | | | | | | |
| alpha-syn | pre-synaptic protein | Abcam | Rabbit Monoclonal | ab138501 | GR221666-8 | 1:400 |
| p-alpha-syn | a-syn phosphorylated at serine 129 | WAKO | Mouse Monoclonal | 015–25191 | LKM6409 | 1:3200 |
| tau | microtubule-associated protein | Abcam | Rabbit Monoclonal | AB32057 | YI062004DS2 | 1:250 |
| p-tau | tau phosphorylated at Serine 202 and Threonine 205 (AT8) | Thermo Fisher | Mouse Monoclonal | MN1020 | QC210802 | 1:200 |
| Immunofluorescence | | | | | | |
| alpha-syn | pre-synaptic protein | Abcam | Rabbit Monoclonal | ab138501 | GR221666-8 | 1:200 |
| p-alpha-syn | a-syn phosphorylated at serine 129 | WAKO | Mouse Monoclonal | 015–25191 | LKM6409 | 1:400 |
| p-alpha-syn | a-syn phosphorylated at serine 129 | Abcam | Rabbit Monoclonal | ab168381 | GR125114-5 | 1:50 |
| tau | microtubule-associated protein | Thermo Fisher | Mouse Monoclonal | MN1000 | UL2886031Z | 1:100 |
| p-tau | tau phosphorylated at Serine 202 and Threonine 205 (AT8) | Thermo Fisher | Mouse Monoclonal | MN1020 | QC210802 | 1:100 |
| PGP 9.5 | panneuronal soma and processes | Invitrogen | Chicken Polyclonal | PA1-10011 | VJ3103801 | 1:500 |

divided into 4 equal size quadrants for analysis to ensure equal sampling throughout each tissue sample. The thickness of the muscularis externa and mucosa were measured per quadrant using the length function. The number of Peyer's patches, round lymphoid follicles within the mucosa of each tissue sample were identified and manually counted. The area of each Peyer's patch was calculated by drawing an ROI around the structure and applying the area function. The anatomical measures of marmoset and rhesus were corrected per body size by creating a ratio of the actual measure divided the actual weight of each individual subject. Because human samples were de-identified by the tissue bank, the correction was made based on typical female-male healthy body weight (average of 50–100 kg = 75 kg).

## Quantification of α-syn, p-α-syn, tau and p-tau expression

α-Syn, p-α-syn, tau and p-tau immunoreactivity (-ir) were calculated separately for the outer longitudinal muscular layer, inner circular muscular layer, mucosa and myenteric ganglia using the percent area above threshold (%AAT) and optical density (OD) functions of the NIH ImageJ software version 2.1.0. %AAT is the percentage of the pixels in the ROI that have a mean grey value above a predetermined threshold and mean OD is the mean grey value, or darkness of all pixels in the ROI.

Images of each layer per quadrant were collected using a Zeiss Axioimager M2 microscope at 10x magnification. Images of the ganglia in the muscularis externa were captured at 40x. ImageJ was calibrated using a step tablet and greyscale values were converted to OD units using the Rodbard function. An optimal OD and %AAT threshold was first calculated per subject for each immunostaining and anatomical area analyzed, and averaged across species. ROIs were drawn for each layer excluding the ganglia and the mean OD and %AAT were measured using a threshold of 0.45 for α-syn-ir and 0.15 for tau-ir. The ganglia of the muscularis externa ROIs were outlined in the 40x images and mean OD and %AAT were measured using a threshold of 0.51 for α-syn-ir, 0.15 for p-α-syn-ir, 0.42 for tau-ir and 0.06 for p-tau-ir.

## Statistical analysis

Data collection and analysis were performed by an investigator blind to the treatment groups. GraphPad Prism (version 9.0, GraphPad Software) was used for statistical analysis. A p value <0.05 was accepted as significant. All data analyzed and reported is listed in S1 Table.

One-way ANOVA followed by post hoc Bonferroni multiple comparisons was used to compare the following measures between species: 1) appendix/cecum perimeter, 2) appendix/cecum diameter, 3) GI layer thickness, 4) number of Peyer's patches within the mucosa, 5) area of Peyer's patches within the mucosa, 6) OD and 7) %AAT of protein expression in the muscularis externa ganglia. Protein expression in the outer longitudinal, inner circular and mucosa layer was compared between species using a two-way repeated measures ANOVA as 3 [(Species: Human + Common Marmoset + Rhesus Macaque) × 3 (Level: Outer longitudinal, Inner circular, Mucosa)] followed by post hoc Bonferroni multiple comparisons. P-values for repeated measures were adjusted using a Huynh-Feldt correction to account for potential violations of the sphericity assumption.

## Results

### The appendix and cecum have comparable microanatomical organization across primate species

Analysis of the appendix/cecum confirmed that human, common marmoset and rhesus macaque have overall similar microanatomy (Fig 1A–1C). The three species had mucosa,

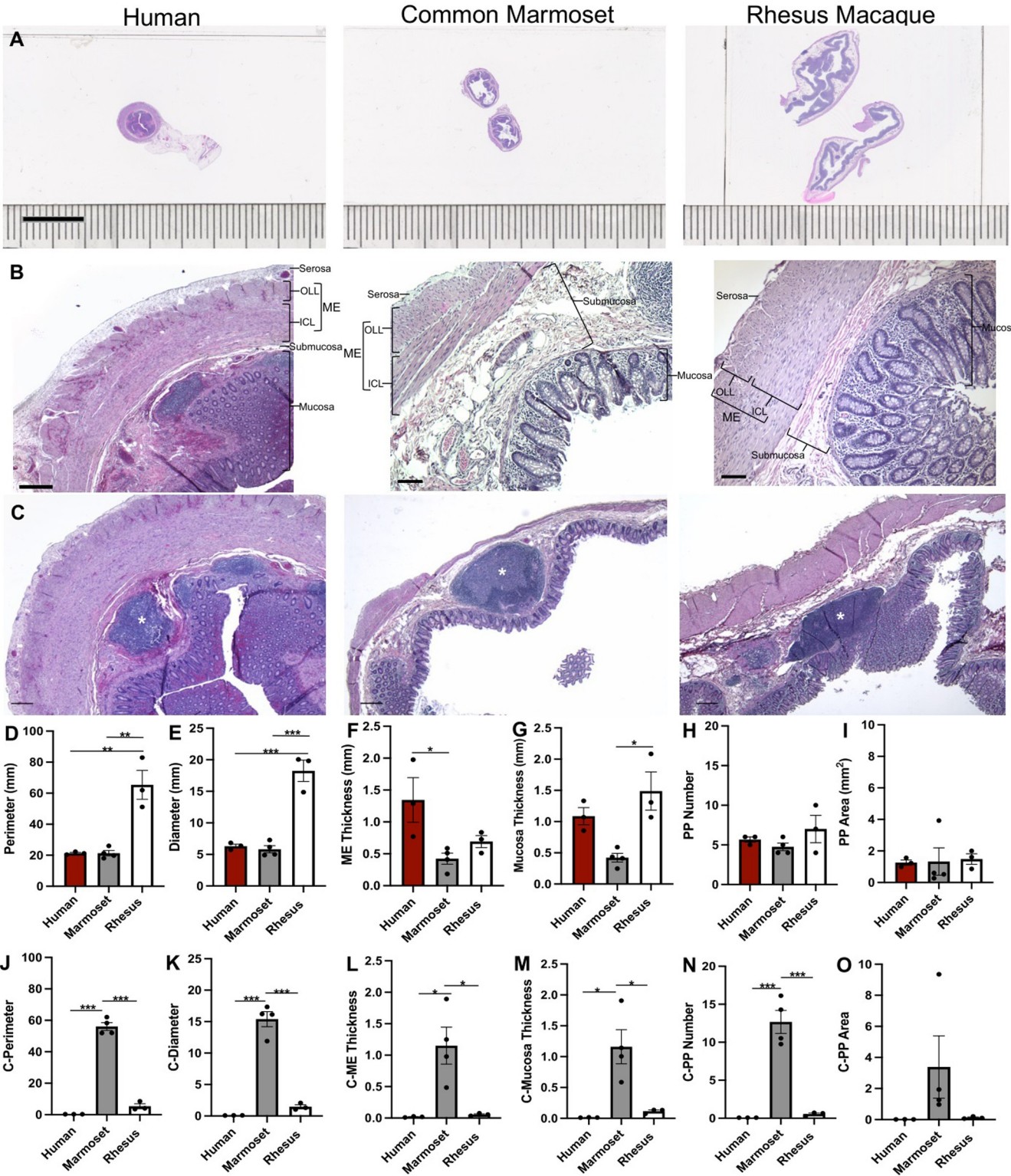

**Fig 1. Organization and dimensions of the appendix/cecum across species.** (A) Images of human appendix and common marmoset and rhesus macaque cecum tissue sections stained with H&E highlighting the difference in size across species, (B) anatomical regions and (C) Peyer's patches (asterisk). Graphs (mean ±SEM) of the (D) perimeter and (E) diameter of the appendix/cecum, (F) muscularis externa thickness, (G) mucosa thickness, (H) number of Peyer's patches and (I) area of Peyer's patches across species. Graphs (J) to (O) correspond to the measures corrected by body size. Scale bar: a, 10mm; b, human, 500μm; rhesus and common marmoset 100μm; c, 500 μm. Abbreviations: ME, Muscularis Externa; OLL, Outer Longitudinal Layer; ICL, Inner Circular Layer; PP, Peyer's patch; C, corresponds to corrected measures by dividing value by subjects weight in kg. *p<0.05, **p<0.01, ***p<0.001.

submucosa and serosa, as well as a muscularis externa consisting of the outer longitudinal and the inner circular layers. A noted difference in organization was that the myenteric plexus of cecum of both NHP species was regularly present between the two layers of the muscularis externa, while in the human appendix the ganglia were heterogeneously distributed throughout the muscularis externa.

An obvious difference across species was the greater size of the rhesus and marmoset ceca compared to the human appendix in relation to average body weight (adult rhesus 7-18kg; marmosets 0.35–0.45kg; human 50-100kg). The cecum perimeter and diameter of the rhesus macaque were significantly larger than in the human appendix (perimeter p = 0.0016; diameter p = 0.0002) and common marmoset cecum (perimeter p = 0.0011; diameter p = 0.0001) (Fig 1A, 1D and 1E). When corrected by body size the common marmoset cecum perimeter and diameter were significantly larger than the human appendix (perimeter p<0.0001; diameter p<0.0001) and rhesus macaque cecum (perimeter p<0.0001; diameter p<0.0001) (Fig 1A, 1J and 1K).

The rhesus macaque mucosa thickness was also significantly larger than the marmoset cecum (p = 0.0106); however, mucosal thickness was not significantly different between the human appendix and ceca of both NHPs (Fig 1G). There was no significant difference in muscularis externa thickness between the rhesus macaque cecum and human appendix and the rhesus macaque cecum and common marmoset cecum, but the human appendix muscularis externa thickness was significantly greater than in the marmoset cecum (p = 0.0341) (Fig 1F). When corrected by body size, mucosal thickness in the common marmoset cecum was significantly larger than in the human appendix (p = 0.0129) and rhesus macaque cecum (p = 0.0205) (Fig 1M). The muscularis externa thickness, when corrected by body size, was significantly larger in the common marmoset cecum than in the human appendix (p = 0.0186) and rhesus macaque cecum (p = 0.0200). (Fig 1L).

Peyer's patches appeared as round lymphoid follicles asymmetrically located throughout the mucosa of the appendix and cecum across species. The number and area of Peyer's patches were not significantly different across the three species (Fig 1C, 1H and 1I). When corrected by body size, the number of Peyer's Patches was significantly larger in the common marmoset cecum than in the human appendix (p = 0.0002) and rhesus macaque cecum (p = 0.0003) (Fig 1N). The area of Peyer's Patches was still not significantly different across species when corrected by body size (Fig 1O).

## α-syn is present in the ENS of the appendix/cecum of human, common marmoset and rhesus

α-Syn-ir in the appendix/cecum of the three species was strikingly abundant and with similar anatomical distribution (Fig 2). Granular α-syn expression was present in the mucosa and muscularis externa, mainly following nerve fibers of all tissue samples (Fig 2A and 2B). In the myenteric ganglia, α-syn-ir was found as diffuse shading and darker punctate spots (Fig 2C).

Quantification of α-syn-ir in the outer longitudinal, inner circular and mucosa layers (excluding ganglia) identified significantly greater %AAT (but not OD) in the human appendix than in the common marmoset and rhesus macaque ceca (overall species x GI layer effect p = 0.0414) (Fig 2D and 2E). In the myenteric ganglia, %AAT and OD of α-syn-ir were not significantly different between species (Fig 2F and 2G).

P-α-syn-ir was present in the appendix/cecum as diffuse staining with very light, punctate spots in the ganglia of the muscularis externa (Fig 3A). Quantification of %AAT and OD of p-α-syn-ir in the ganglia did not show any significant differences in immunoreactivity between species (Fig 3C and 3D).

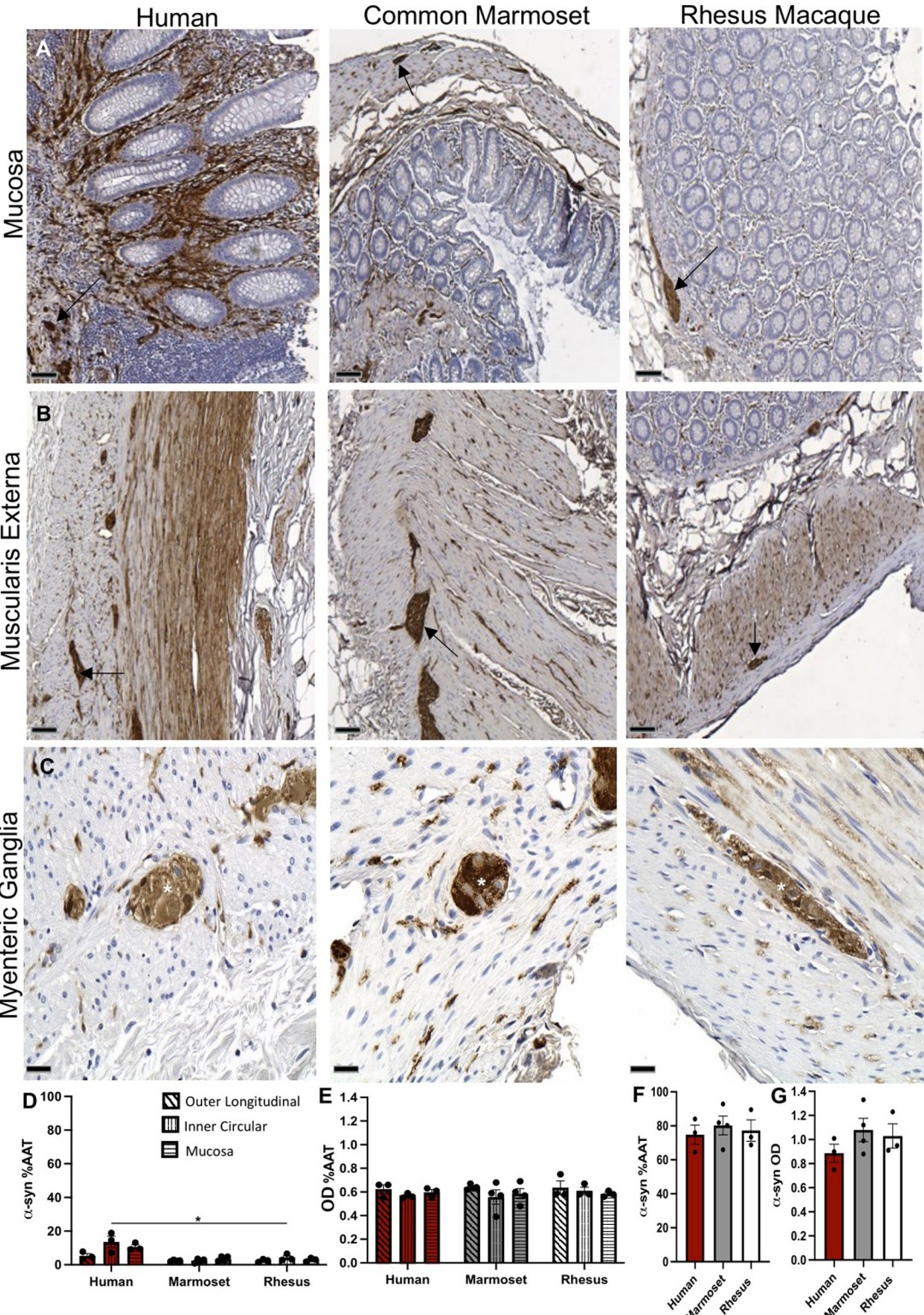

**Fig 2. α-syn expression in the appendix/cecum across species.** Images of α-syn-immunoreactivity (-ir) (brown) in the (A) mucosa (arrows), (B) muscularis externa (arrows) and (C) myenteric ganglia (asterisks) from human appendix, common marmoset and rhesus ceca tissue sections counterstained with hematoxylin (blue). Graphs (mean ±SEM) of α-syn-ir (D,E) in gastrointestinal layers and (F,G) myenteric ganglia across species. Scale bar; a, 100μm; b, 100μm; c, 25 μm. Abbreviations: α-syn, alpha-synuclein; %AAT, percent area above threshold; OD, optical density. *p<0.05.

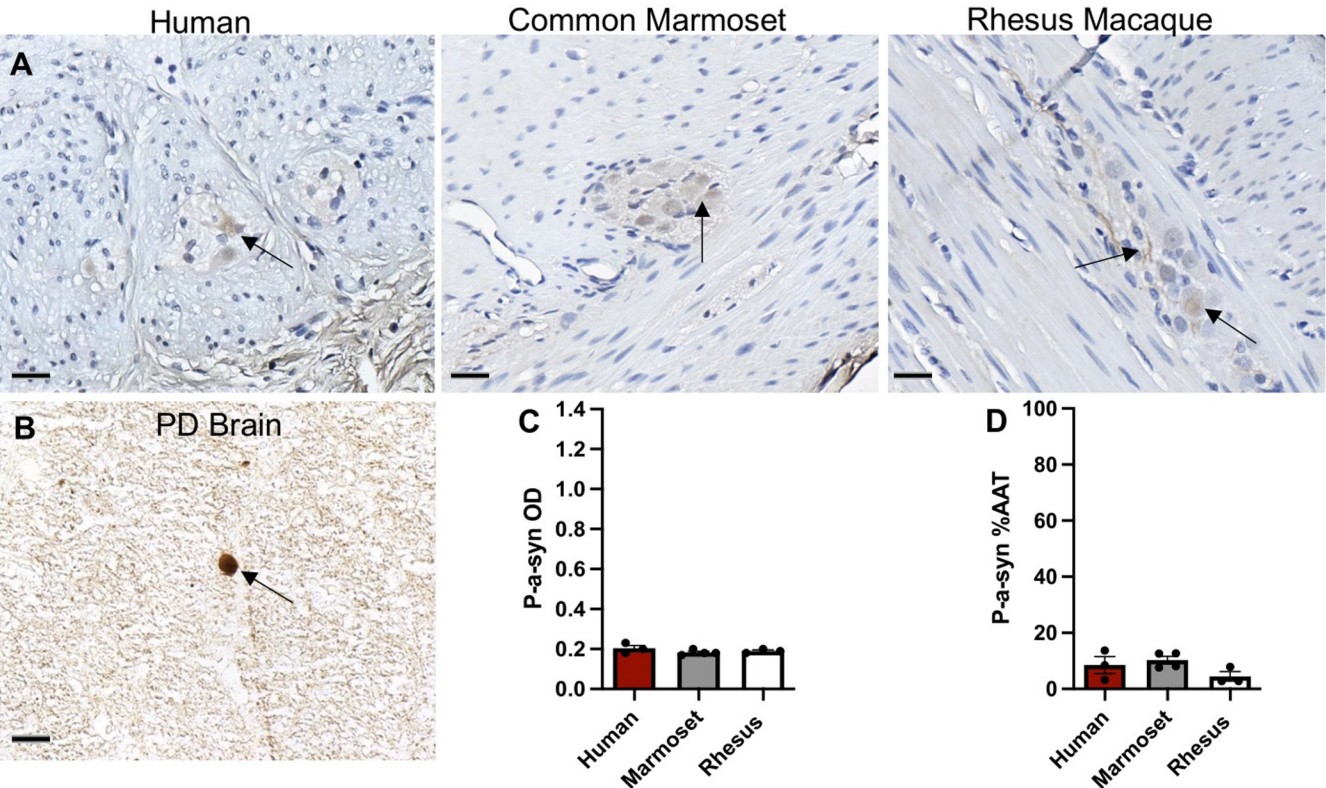

**Fig 3. P-α-syn expression in the appendix/cecum across species.** (A) Images of p-α-syn-immunoreactivity (-ir) (brown) in the myenteric ganglia (arrows) in human appendix, common marmoset and rhesus macaque ceca tissue sections counterstained with hematoxylin (blue). (B) Image of p-α-syn-ir in a Lewy body from human Parkinson's disease brain tissue section. Graphs (mean ±SEM) of p-α-syn-ir (C, D) in muscularis externa ganglia across species. Scale bar; 25 μm. Abbreviations: %AAT, percent area above threshold; OD, optical density; PD, Parkinson's disease. p-α-syn, serine 129 phosphorylated alpha-synuclein.

## Tau expression in the ENS of the appendix/cecum matches α-syn-ir

Across all three species, tau-ir in the appendix/cecum was abundantly present within the myenteric ganglia and nerve fibers in the mucosa and muscularis externa, following a similar anatomical distribution to α-syn-ir (Fig 4A–4C). Along nerve fibers, tau expression was granular and, in myenteric ganglia, was present as diffuse shading as well as dark punctate spots. % AAT and OD of tau-ir in myenteric ganglia were not significantly different between species (Fig 4F and 4G). Across GI layers, %AAT of tau-ir (but not OD) was significantly larger in the human appendix than in the common marmoset and rhesus macaque cecum (overall species x GI layer effect p = 0.0342) (Fig 4D and 4E).

Similar to p-α-syn-ir, scattered p-tau-ir was present within myenteric ganglia across all three species (Fig 5A) and no significant differences were identified in %AAT and OD of p-tau-ir in ganglia (Fig 5C and 5D).

## The ENS ganglia neurons in the appendix/cecum express α-syn and tau

Triple labeling immunofluorescence confirmed that α-syn and tau were expressed in PGP9.5 positive myenteric ganglia and nerve fibers in the muscularis externa and mucosa. Within the myenteric ganglia, α-syn-ir colocalized with p-α-syn-ir and PGP9.5-ir (Fig 6A) and α-syn-ir colocalized with tau and PGP9.5 (Fig 6B). We also found colocalization of p-α-syn, p-tau and PGP 9.5 within the myenteric ganglia (Fig 6C).

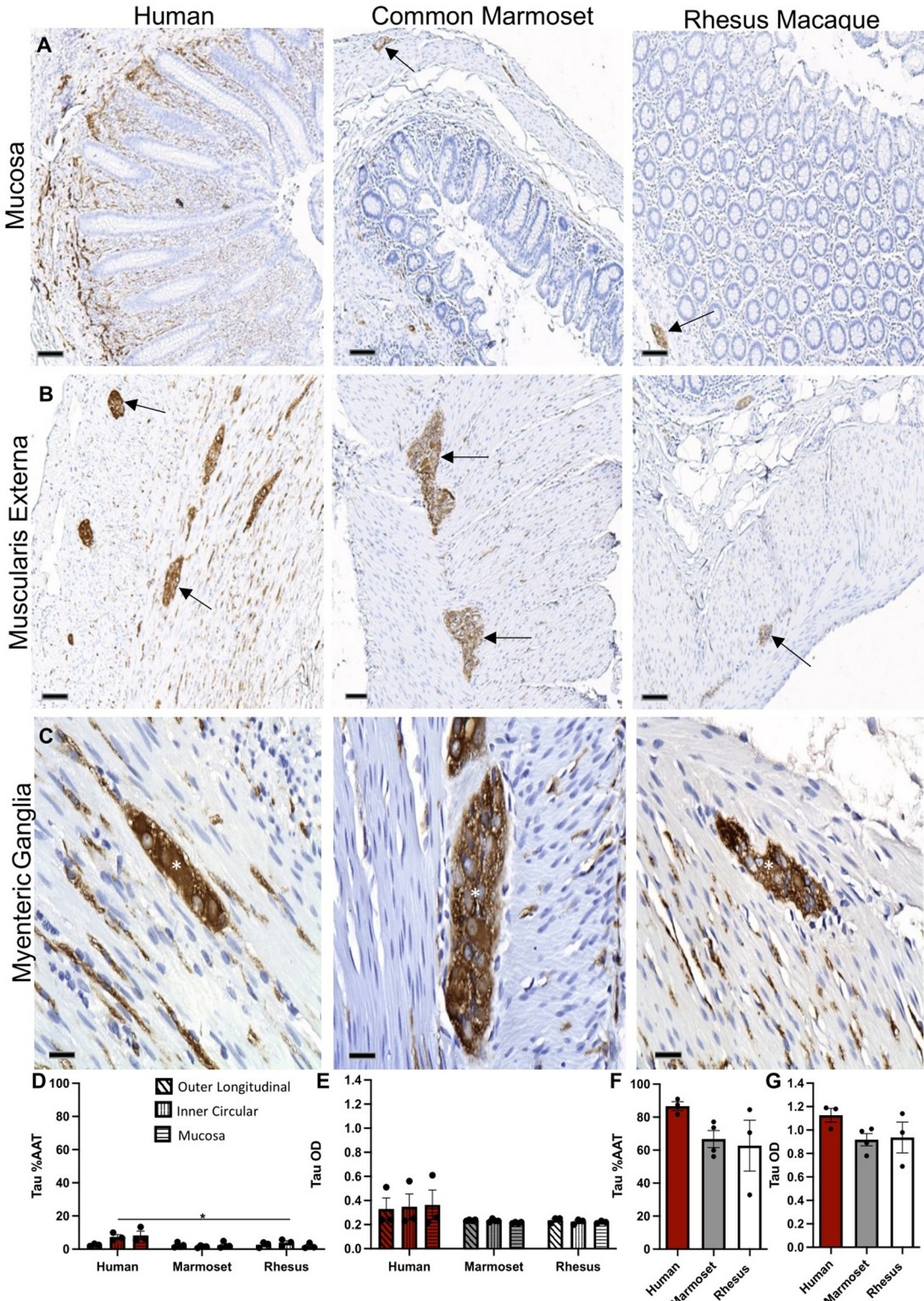

**Fig 4. Tau expression in the appendix/cecum across species.** Images of tau-immunoreactivity (-ir) (brown) in the (A) mucosa (arrows), (B) muscularis externa (arrows) and (C) myenteric ganglia (asterisks) from human appendix, common marmoset and rhesus ceca tissue sections counterstained with hematoxylin (blue). Graphs (mean ±SEM) of tau-ir (D,E) in gastrointestinal layers and (F,G) myenteric ganglia across species. Scale bar; a, 100μm; b, 100μm; c, 25 μm. Abbreviations: %AAT, percent area above threshold; OD, optical density. *p<0.05.

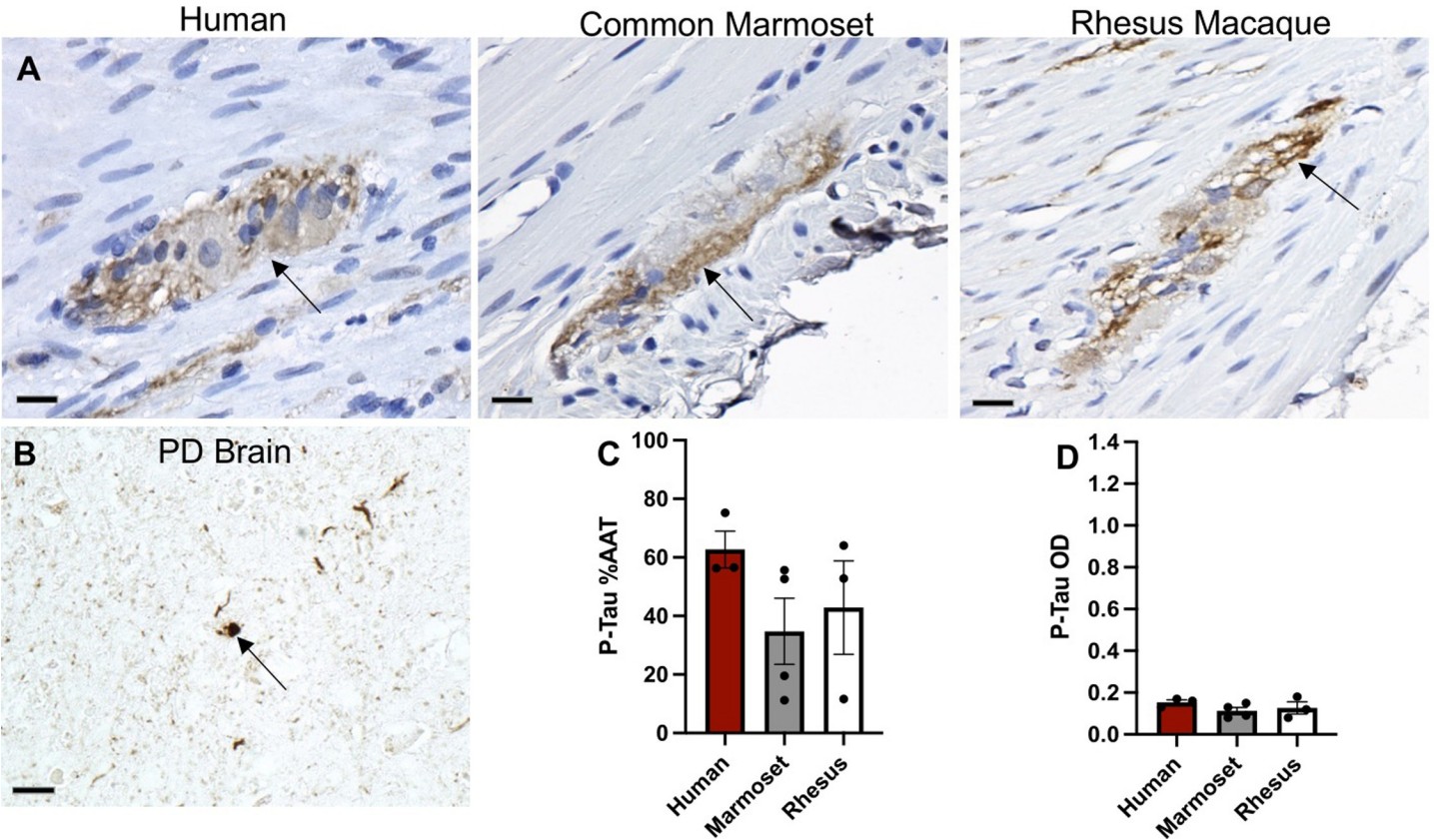

**Fig 5. P-tau expression in the appendix/cecum across species.** (A) Images of p-tau-immunoreactivity (-ir) (brown) in the myenteric ganglia (arrows) in human appendix, common marmoset and rhesus macaque ceca tissue sections counterstained with hematoxylin (blue). (B) Image of p-tau-ir in a Lewy body from human Parkinson's disease brain tissue section. Graphs (mean ±SEM) of p-tau-ir (C, D) in muscularis externa ganglia across species. Scale bar; 25 μm. Abbreviations: %AAT, percent area above threshold; OD, optical density; PD, Parkinson's disease; p-tau, phosphorylated tau.

## Discussion

Our results demonstrate that the appendix of humans and the cecum of common marmosets and rhesus macaques share comparable microanatomy as well as similar expression of the neuronal proteins α-syn, p-α-syn, tau and p-tau in the myenteric ganglia. To the best of our knowledge, this is the first report of α-syn, p-α-syn, tau and p-tau expression outside the CNS for NHPs. Additionally, we are the first to report tau and p-tau expression in the human appendix.

The defining characteristics of the human vermiform appendix are its thick apical walls, location at the end of the cecum and presence of dense amounts of lymphoid tissue in the lamina propria of the mucosa and in the submucosa [1, 2]. The lack of a "typical" appendix in animals, like rhesus macaques and common marmosets, has puzzled scientists for over 100 years. In 1900, Berry analyzed the ceca of thirty-one different species, including the common marmoset and compared their cecum to the human cecum and vermiform appendix [23]. He found that in animals lacking an appendix, the cecum contained increased lymphoid tissue compared to other parts of the large intestine and ultimately concluded that the cecum of animals lacking an appendix is histologically equivalent to the human appendix [23]. One hundred years later, Fisher echoed similar sentiments regarding the lack of comprehensive literature on the cecum of animals without an appendix [1]. After analyzing the sparse literature on 60 NHP species, she reported that lymphoid tissue was present in the cecal apex of

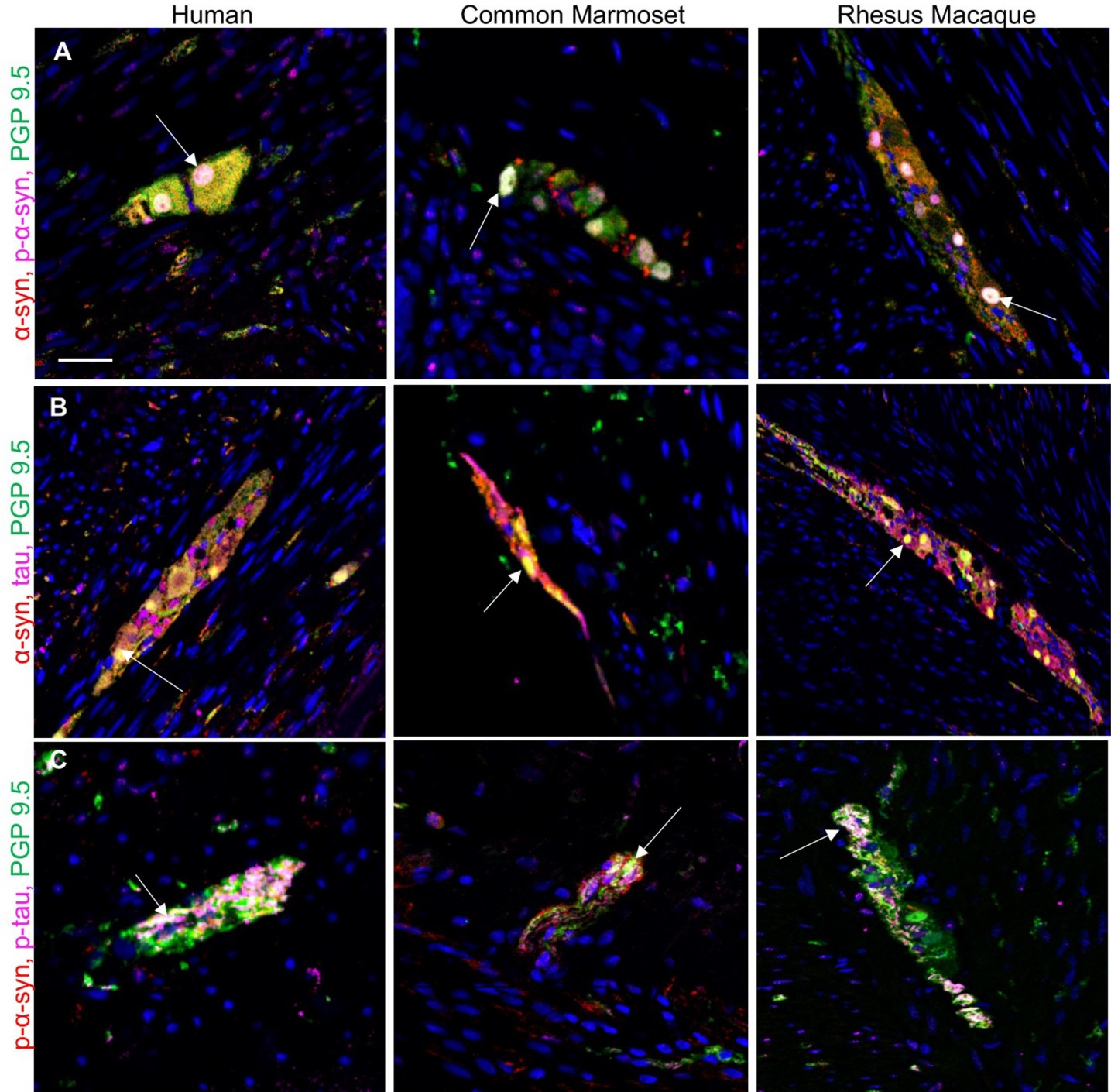

**Fig 6. Representative photomicrographs showing immunofluorescent co-labeling.** (A) α-syn, p-α-syn and PGP9.5 (B) α-syn, tau and PGP9.5 (C) p-α-syn, p-tau and PGP9.5, in myenteric ganglia in human appendix, common marmoset cecum and rhesus macaque cecum. Abbreviations: p-α-syn, serine 129 phosphorylated alpha-synuclein; p-tau, phosphorylated tau; PGP9.5, protein gene product 9.5. Scale bar: 50 μm.

most of NHPs, including macaques and marmosets [1]. Our findings confirmed the presence of Peyer's patches within the mucosa of human appendix and the cecum of marmosets and rhesus. The lymphoid follicles had similar area across species, but their number was greater in

common marmoset cecum than rhesus cecum and human appendix when corrected by body size. Additionally, the thickness of the mucosa layer was significantly greater in marmosets, further suggesting a greater amount of lymphoid tissue in the marmoset cecum.

The neuronal protein α-syn was present across all species in the myenteric ganglia and along the neuronal axons of muscularis externa and mucosa. Quantification of α-syn-ir in ganglia of the muscularis externa showed no significant difference across species, yet α-syn-ir % AAT was significantly greater in human GI layers than in NHPs. A possible explanation for this finding could be the heterogeneous distribution of myenteric ganglia throughout the layers of the muscularis externa in the human appendix, compared to the orderly ganglia of the NHP cecum, which could contribute a greater number of nerve fibers emanating from the ganglia and crisscrossing the muscularis externa.

In our study, p-α-syn-ir was present in the ganglia and nerve fibers of the human appendix and marmoset and rhesus cecum. Like α-syn-ir, it was observed as diffuse shading and sporadic punctate spots, except lighter and more scattered. P-α-syn-ir is prevalent in the GI tract of healthy and PD patients including esophagus, stomach, small intestine, colon and appendix [19]. In appendices from healthy patients and patients with PD, p-α-syn has been reported to be present within the nerve fibers along the muscularis externa and mucosa as well as in submucosa ganglia and myenteric ganglia [19]. P-α-syn-ir was expressed in 67% (2/3) of appendix samples from PD patients and in 75% (6/8) of control appendix samples [19]. Proteinase K-resistant α-syn aggregates have been found in the mucosa, myenteric ganglia and nerve fibers of the vermiform appendix in both healthy and PD patients [4].

Similar to α-syn-ir, tau-ir was present in the myenteric ganglia and neuronal axons of muscularis externa and mucosa of the human appendix and NHP ceca. Few publications have reported on the expression of tau outside the central nervous system in humans and non-human species. Our findings match available descriptions of tau-ir in the myenteric plexus of the colon in mice and humans [17], as well as along nerve fibers in the muscularis externa and mucosa of the human colon [15, 17]. Interestingly, while tau-ir was most prominent in the colon in humans, it was also present in submandibular gland tissue samples [24].

Literature on p-tau immunohistochemistry in the ENS is limited and the few reports available have used antibodies targeting tau phosphorylation at different amino acids. In this study, we identified phosphorylated tau-ir at serine 202 and threonine 205 (pSer202/Thr205-tau) as diffuse shading within the myenteric ganglia across all species. Although there are no reports in appendix or cecum, pSer202/Thr205-tau-ir was described in healthy human colon samples and in the myenteric ganglia of the colon and ileum of rats [24, 25]. Other studies found expression of pSer396-tau in the colonic myenteric ganglia of healthy and Crohn's disease patients [17, 26].

Although it is well accepted that GI symptoms may precede the onset of sporadic PD for many years, the evidence linking PD onset and progression to local ENS proteinopathy is mixed [13, 27, 28]. LBs have been found in the myenteric and Meissner's plexus of the enteric nervous system [29], yet new controlled studies showed that α-syn and p-α-syn (the basic building blocks for LBs) were similarly expressed in the colonic mucosa of healthy and PD patients [30, 31]. LBs composed of insoluble aggregated, detergent and proteinase K-resistant a-synuclein have been injected into the striatum and the stomach and duodenum ventral wall of baboons [32]. Both striatal and ENS injections induced α-syn pathology in the ENS, indicating a bidirectional propagation of α-syn pathology in NHPs [32]. Further analysis of the appendix may provide clues on PD etiology, as appendectomy has been linked to a lower risk of PD and a delayed onset of PD [3, 4]. The greater abundance of α-syn-ir fibers in the vermiform appendix compared to other GI sections suggests the appendix as a reservoir for the initiation of enteric α-syn aggregation [18]; perhaps changes in α-syn-ir follow a temporal pattern associated to specific triggers and/or conditions.

Inflammation has been proposed to facilitate α-syn phosphorylation and aggregation [33, 34]. Many GI pathologies characterized by chronic inflammation, including ulcerative colitis and Crohn's disease, have been linked to higher risk of PD [35]. Moreover, patients with Crohn's and other autoimmune, inflammatory diseases have genetic variants in common with some PD patients [36, 37]. Like α-syn, tau also accumulates in the colon of patients with Crohn's disease [26]. We have found that chronic colitis in common marmosets increases expression of p-α-syn in the colonic myenteric ganglia [22]. Yet, this change in p-α-syn was not present in cynomolgous 2 weeks after acute colitis triggered by oral exposure to the food borne pathogen listeria suggesting that a protracted inflammatory state is needed to elicit proteinopathy [38]. The appendix, in addition to the greater amount of α-syn, houses substantial amounts of GALT. The decreased risk of developing PD following appendectomy could be related to decreased inflammation; appendectomy has also been linked to decreased risk of developing ulcerative colitis [39] which, in turn, increases the risk of PD by 30% [35]. Interestingly, an imaging study of the appendix of PD patients (n = 100) found that half (n = 53) had chronic appendicitis [40]. The pathological report of the resected appendix in a subset of the subjects that underwent appendectomy (n = 7) described the presence of inflammation as well as accumulation of α-syn [40]. The GI microbiome, which has been found altered in PD patients [41], may also have a role in the onset of PD. In that regard, the appendix is proposed to be a safe house for commensal bacteria, which can repopulate the gut in the event of diarrhea, a common symptom of irritable bowel disease that has also been linked to increase risk for PD [2, 41, 42].

In addition to prospective studies targeting the appendix and GI in human populations at risk of PD, experiments in animal models can provide insight into specific issues that could benefit from controlled conditions and/or systematic in vivo sampling followed by postmortem analysis. The selection of the species will need to be carefully considered as differences in the GI of large versus small mammals may affect the results. ENS structure in humans and larger mammals such as NHPs, is more complex with multiple distinct neural networks forming the submucosa plexus mammals [43–45] as opposed to the singular submucosa network seen in smaller mammals [28, 46]. Understanding differences across species can provide insight on the advantages and limitations of the models and their validity for clinical translation.

## Conclusions

This study provides critical translational evidence that the cecum of the common marmoset and the rhesus macaque is remarkably similar to the human appendix. They share comparable microanatomy and immunoreactivity of α-syn, p-α-syn, tau and p-tau. To our knowledge, this is the first description of tau and p-tau in the human appendix, as well as the common marmoset and rhesus macaque ceca. Further studies on the role of the gastrointestinal tract, proteinopathy and PD onset are needed. Based on our results we propose that rhesus macaques and common marmosets are suitable as model species for studying the development of PD linked to α-syn and tau pathological changes in the ENS.

## Supporting information

**S1 Table. Dataset used for statistical analysis.** The following tables are included: (A) Appendix and Cecum perimeter, (B) Appendix and Cecum diameter, (C) GI layer thickness, (D) Number for Peyer's Patches, (E) Area of Peyer's Patches, (F) α-Syn GI Layer OD and %AAT, (G) α-Syn Ganglia OD and %AAT, (H) P-α-syn Ganglia OD and %AAT, (I) Tau GI Layer OD and %AAT, (J) Tau Ganglia OD and %AAT, (K) P-tau Ganglia OD and %AAT. (XLSX)

## Acknowledgments

The authors gratefully acknowledge Robert Becker for imaging support, the dedicated animal care and veterinary staff at the Wisconsin National Primate Research Center for their technical support, the University of Wisconsin Translational Research Initiatives in Pathology laboratory (TRIP), supported by the UW Department of Pathology and Laboratory Medicine, and the University of Wisconsin Carbone Cancer Center (UWCCC) (P30 CA014520) for use of its services.

## Author Contributions

**Conceptualization:** Alexandra D. Zinnen, Marina E. Emborg.

**Data curation:** Alexandra D. Zinnen, Marina E. Emborg.

**Formal analysis:** Alexandra D. Zinnen, Jeanette M. Metzger.

**Funding acquisition:** Alexandra D. Zinnen, Marina E. Emborg.

**Investigation:** Alexandra D. Zinnen, Jonathan Vichich, Jeanette M. Metzger, Julia C. Gambardella, Viktoriya Bondarenko, Heather A. Simmons, Marina E. Emborg.

**Methodology:** Alexandra D. Zinnen, Jeanette M. Metzger, Viktoriya Bondarenko, Heather A. Simmons, Marina E. Emborg.

**Project administration:** Alexandra D. Zinnen, Marina E. Emborg.

**Resources:** Marina E. Emborg.

**Software:** Alexandra D. Zinnen.

**Supervision:** Marina E. Emborg.

**Validation:** Alexandra D. Zinnen, Jeanette M. Metzger, Viktoriya Bondarenko, Heather A. Simmons, Marina E. Emborg.

**Visualization:** Alexandra D. Zinnen.

**Writing – original draft:** Alexandra D. Zinnen, Marina E. Emborg.

**Writing – review & editing:** Alexandra D. Zinnen, Jonathan Vichich, Jeanette M. Metzger, Julia C. Gambardella, Viktoriya Bondarenko, Heather A. Simmons, Marina E. Emborg.

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
