## [Decision Letter · Decision Letter 0]

9 Feb 2022

PONE-D-21-40458Alpha-synuclein and tau are abundantly expressed in the ENS of the human appendix and monkey cecumPLOS ONE

Dear Dr. Emborg,

Thank you for submitting your manuscript to PLOS ONE. After careful consideration, we feel that it has merit but does not fully meet PLOS ONE’s publication criteria as it currently stands. Therefore, we invite you to submit a revised version of the manuscript that addresses the points raised during the review process.

Reviewer #2's comment--"The work presented here is novel and the manuscript is well-written. The experimental design can, however, be improved" is what has driven this decision. The main criticism is the small sample size; this was also pointed out by Reviewer #1. Please address the four bullet points raised by Reviewer #2. These points have to do with sample size, statistics, body weights, figures and references. Please also address the minor points raised by Reviewer#1. Addressing these various points will improve your manuscript. 

We look forward to receiving your revised manuscript.

Kind regards,

Stephan N. Witt, Ph.D.

Academic Editor

PLOS ONE

Journal Requirements:

4. Thank you for stating the following in the Funding Section of your manuscript: 

"This research was funded by grants from the National Institute of Health [NIH P51OD011106] (ME), the Parkinson’s Foundation [PF-APDA-SFW-1922] (AZ) and Trewartha Senior Honors Thesis Research Award from the University of Wisconsin–Madison (AZ). The Funders had no role in study design, data collection and analysis, decision to publish, or preparation of the manuscript. "

"This research was funded by grants from the National Institute of Health [NIH P51OD011106; https://orip.nih.gov/] (ME), the Parkinson’s Foundation [PF-APDA-SFW-1922; https://www.parkinson.org/] (AZ) and Trewartha Senior Honors Thesis Research Award from the University of Wisconsin–Madison [https://wisc.academicworks.com/opportunities/48209] (AZ). The funders had no role in study design, data collection and analysis, decision to publish, or preparation of the manuscript."

Reviewers' comments:

Reviewer's Responses to Questions

**Comments to the Author**

1. Is the manuscript technically sound, and do the data support the conclusions?

Reviewer #1: Yes

Reviewer #2: No

2. Has the statistical analysis been performed appropriately and rigorously? 

Reviewer #1: Yes

Reviewer #2: No

3. Have the authors made all data underlying the findings in their manuscript fully available?

Reviewer #1: Yes

Reviewer #2: Yes

4. Is the manuscript presented in an intelligible fashion and written in standard English?

Reviewer #1: Yes

Reviewer #2: Yes

5. Review Comments to the Author

Reviewer #1: Review report for the manuscript entitled "Alpha-synuclein and tau are abundantly expressed in the ENS of the human appendix and monkey cecum"

This is a prospective study that proposes to use non-human primate species such as common marmosets and monkeys as a model system for gastrointestinal proteinopathy mediated Parkinson’s Disease pathology.

The following are the key observations in this study

1. The caeca of common marmosets and monkeys shows comparable anatomical organization to the human appendix.

2. The ENS of marmosets and monkeys expresses α-syn, phosphorylated α-syn, tau and phosphorylated tau

3. The authors have carefully analyzed the expression patterns of α-syn, phosphorylated α-syn, tau and phosphorylated tau in the enteric nervous system ganglia of muscularis externa, submucosa, and mucosal regions in human appendix tissue and NHP’s caecum

The manuscript is well constructed, and the experiments were properly designed, and the analysis was carefully performed (except n=2 for human appendix tissue). After considering the originality, quality of the results, constraints in using human tissue in research and the possibility of utilizing a non-human primate as a model system to study Gut-Brain axis in PD pathology the manuscript can be considered for publication in PLOS ONE despite n=2 for human appendix.

Minor suggestion:

1. In line 409-412: “The decreased risk of developing PD following appendectomy could be related to decreased inflammation; appendectomy has also been linked to decreased risk of developing ulcerative colitis which, in turn, increases the risk of PD”. The authors should comment on this.

2. Line 427: ‘provides’ repeated twice

Reviewer #2: The manuscript from Zinnen et al. aims to compare the human appendix with its non-human primate counterpart, the cecum, found in common marmosets and rhesus macaques. The authors make a direct comparison in terms of microanatomy and analyze expression of PD-related proteins alpha-synuclein and tau in the enteric nervous system of each. The work presented here is novel and the manuscript is well-written. The experimental design can, however, be improved. Below is a list of concerns regarding the study:

• The sample size for each group is too small, especially for the human samples. The statistical analyses are not appropriate with n=2 and one cannot make strong statements about the general applicability of the findings. While it can be difficult to obtain tissue samples during the pandemic, n=2 is not adequate for statistical analyses. Furthermore, the investigators only require normal human appendix tissue, and it is not uncommon that patients undergoing routine appendectomy are found to have normal (not inflamed) appendix. Therefore, it cannot be difficult to obtain at least 2-3 more samples which would allow the study to address rigor and reproducibility in the group of human tissues.

• In the Materials and Methods section, the authors state that typical female-male healthy body weight (average of 50-100 kg= 75 kg) was used for calculations, but in the Results section, under the sub-heading ‘The appendix and cecum have comparable microanatomical organization across primate species’, the average weight range is stated as 50-120 kg. Would it be possible to retrieve the real values (instead of a range) from the biobank without compromising the identities of the tissue donors?

• In Figs 2 and 4, panel D, no symbols are used to denote significance levels. I would suggest the authors add this to the images to improve readability.

• When discussing the relevance of appendix tissue to Parkinson disease, the authors should cite two more recent papers that are highly relevant. They are PMID: 33876851 and PMID: 32595591

6. PLOS authors have the option to publish the peer review history of their article (what does this mean?). If published, this will include your full peer review and any attached files.

Reviewer #1: No

Reviewer #2: No

---

## [Author Response · Author response to Decision Letter 0]

6 Apr 2022

Response to Reviewers

Below please find the full reviewer report with the reviewer’s comments italicized and the authors’ responses to the comments in bold. NRN (no response needed) was used when an author’s response was not needed. 

Reviewer #1

1. Is the manuscript technically sound, and do the data support the conclusions?

Reviewer #1: Yes

AU: NRN 

2. Has the statistical analysis been performed appropriately and rigorously? 

Reviewer #1: Yes

AU: NRN

3. Have the authors made all data underlying the findings in their manuscript fully available?

Reviewer #1: Yes

AU: NRN 

4. Is the manuscript presented in an intelligible fashion and written in standard English?

Reviewer #1: Yes

AU: NRN 

5. Review #1 Comments to the Author

This is a prospective study that proposes to use non-human primate species such as common marmosets and monkeys as a model system for gastrointestinal proteinopathy mediated Parkinson’s Disease pathology.

The following are the key observations in this study

1. The caeca of common marmosets and monkeys shows comparable anatomical organization to the human appendix.

2. The ENS of marmosets and monkeys expresses α-syn, phosphorylated α-syn, tau and phosphorylated tau

3. The authors have carefully analyzed the expression patterns of α-syn, phosphorylated α-syn, tau and phosphorylated tau in the enteric nervous system ganglia of muscularis externa, submucosa, and mucosal regions in human appendix tissue and NHP’s caecum

The manuscript is well constructed, and the experiments were properly designed, and the analysis was carefully performed (except n=2 for human appendix tissue). After considering the originality, quality of the results, constraints in using human tissue in research and the possibility of utilizing a non-human primate as a model system to study Gut-Brain axis in PD pathology the manuscript can be considered for publication in PLOS ONE despite n=2 for human appendix.

AU: We thank the reviewer for acknowledging the value of the project and the constraints in using human tissue in research. For this revised version of the manuscript, we were able to add an additional human cecum, increasing the n of the human sample to 3, which matches the rhesus sample size. 

Reviewer #1 Minor suggestion:

In line 409-412: “The decreased risk of developing PD following appendectomy could be related to decreased inflammation; appendectomy has also been linked to decreased risk of developing ulcerative colitis which, in turn, increases the risk of PD”. The authors should comment on this.

AU: We thank the reviewer for this suggestion. In the discussion section of the revised manuscript we have added wording and additional citations to the mentioned paragraph expanding on this concept.

Line 427: ‘provides’ repeated twice

AU: Acknowledged and corrected. 

6. Do you want your identity to be public for this peer review? 

Reviewer #1: No

AU:NRN

Reviewer # 2 

1. Is the manuscript technically sound, and do the data support the conclusions?

Reviewer #2: No

AU: Based on the reviewer #2’s comments on item 5, we interpreted that the main concern is the small human sample size. For this revised version of the manuscript we were able to add an additional human cecum, increasing the n of the human sample to 3, which matches the rhesus sample size, to strengthens the statistical analysis and conclusions. 

2. Has the statistical analysis been performed appropriately and rigorously? 

Reviewer #2: No

AU: As stated above, we have increased the human appendix sample size to 3, aiming to strengthen the statistical analysis. 

3. Have the authors made all data underlying the findings in their manuscript fully available?

Reviewer #2: Yes

AU: NRN

4. Is the manuscript presented in an intelligible fashion and written in standard English?

Reviewer #2: Yes

AU: NRN

5. Review Comments to the Author

Reviewer #2: The manuscript from Zinnen et al. aims to compare the human appendix with its non-human primate counterpart, the cecum, found in common marmosets and rhesus macaques. The authors make a direct comparison in terms of microanatomy and analyze expression of PD-related proteins alpha-synuclein and tau in the enteric nervous system of each. The work presented here is novel and the manuscript is well-written. The experimental design can, however, be improved. Below is a list of concerns regarding the study

• The sample size for each group is too small, especially for the human samples. The statistical analyses are not appropriate with n=2 and one cannot make strong statements about the general applicability of the findings. While it can be difficult to obtain tissue samples during the pandemic, n=2 is not adequate for statistical analyses. Furthermore, the investigators only require normal human appendix tissue, and it is not uncommon that patients undergoing routine appendectomy are found to have normal (not inflamed) appendix. Therefore, it cannot be difficult to obtain at least 2-3 more samples which would allow the study to address rigor and reproducibility in the group of human tissues.

AU:. Please note that in our experience finding healthy, noninflamed human appendix samples is difficult as it is not commonly collected by tissue banks. In spite of this issue, we were able to add an additional human cecum, increasing the n of the human sample to 3, which matches the rhesus sample size, to strengthens the statistical analysis and conclusions.

• In the Materials and Methods section, the authors state that typical female-male healthy body weight (average of 50-100 kg= 75 kg) was used for calculations, but in the Results section, under the sub-heading ‘The appendix and cecum have comparable microanatomical organization across primate species’, the average weight range is stated as 50-120 kg. Would it be possible to retrieve the real values (instead of a range) from the biobank without compromising the identities of the tissue donors?

AU: Please note that the human appendix tissue samples from this study were fully de-identified, anonymized samples from the Translational Research Initiative (TRIP) Lab in the department of Pathology and Laboratory Medicine at the University of Wisconsin – Madison and the Translational Science Biocore Biobank (TSB Biobank) at the University of Wisconsin Carbone Cancer Center. The TRIP lab and the TSB Biobank will not release the real weight value of the de-identified human tissue donors. In the revised version of the manuscript, we have corrected the average weight range stated in the results to match the correct weight range used for calculations stated in the methods (average of 50-100 kg= 75 kg).

• In Figs 2 and 4, panel D, no symbols are used to denote significance levels. I would suggest the authors add this to the images to improve readability.

AU: We thank the reviewer for the suggestion. In this revised version of the figures, we have added symbols to Figure 2 and 4, panel D, to denote significance levels. 

• When discussing the relevance of appendix tissue to Parkinson disease, the authors should cite two more recent papers that are highly relevant. They are PMID: 33876851 and PMID: 32595591

AU: We thank the reviewer for the recommended publications. We incorporated them in the introduction and discussion section of the revised manuscript. 

6. Do you want your identity to be public for this peer review? 

Reviewer #2: No

AU: NRN

Response to Editor Comments: 

Please address the four bullet points raised by Reviewer #2. These points have to do with sample size, statistics, body weights, figures and references. Please also address the minor points raised by Reviewer#1.

In this revised version of the manuscript we aimed to address reviewer #2's concern by adding an additional human sample and, thus, re-doing graphs and statistics as needed.

We have also addressed Reviewer #1's minor concern adding appropriate wording to the manuscript.

---

## [Decision Letter · Decision Letter 1]

17 May 2022

Alpha-synuclein and tau are abundantly expressed in the ENS of the human appendix and monkey cecum

PONE-D-21-40458R1

Dear Dr. Emborg,

We’re pleased to inform you that your manuscript has been judged scientifically suitable for publication and will be formally accepted for publication once it meets all outstanding technical requirements.

Kind regards,

Stephan N. Witt, Ph.D.

Academic Editor

PLOS ONE

Additional Editor Comments (optional):

Sorry for the delay in making the decision. Some things are out of my control.

Reviewers' comments:

Reviewer's Responses to Questions

**Comments to the Author**

1. If the authors have adequately addressed your comments raised in a previous round of review and you feel that this manuscript is now acceptable for publication, you may indicate that here to bypass the “Comments to the Author” section, enter your conflict of interest statement in the “Confidential to Editor” section, and submit your "Accept" recommendation.

Reviewer #1: All comments have been addressed

2. Is the manuscript technically sound, and do the data support the conclusions?

Reviewer #1: Yes

3. Has the statistical analysis been performed appropriately and rigorously? 

Reviewer #1: Yes

4. Have the authors made all data underlying the findings in their manuscript fully available?

Reviewer #1: Yes

5. Is the manuscript presented in an intelligible fashion and written in standard English?

Reviewer #1: Yes

6. Review Comments to the Author

Reviewer #1: Appreciate the authors for increasing the human sample (n=3). The authors have completely addressed the comments.

7. PLOS authors have the option to publish the peer review history of their article (what does this mean?). If published, this will include your full peer review and any attached files.

Reviewer #1: No

---

## [Editor Report · Acceptance letter]

1 Jun 2022

PONE-D-21-40458R1 

Alpha-synuclein and tau are abundantly expressed in the ENS of the human appendix and monkey cecum 

Dear Dr. Emborg:

I'm pleased to inform you that your manuscript has been deemed suitable for publication in PLOS ONE. Congratulations! Your manuscript is now with our production department. 

Kind regards, 

on behalf of

Dr. Stephan N. Witt 

Academic Editor

PLOS ONE